# Role of Translationally Controlled Tumor Protein (TCTP) in the Development of Hypertension and Related Diseases in Mouse Models

**DOI:** 10.3390/biomedicines10112722

**Published:** 2022-10-27

**Authors:** Jeehye Maeng, Kyunglim Lee

**Affiliations:** Graduate School of Pharmaceutical Sciences, College of Pharmacy, Ewha Womans University, Seoul 03760, Korea

**Keywords:** ApoE KO, apolipoprotein E knockout mice, atherosclerosis, hypertension, Na,K-ATPase, TCTP, TCTP-overexpressing transgenic mice, translationally controlled tumor protein, TCTP-TG

## Abstract

Translationally controlled tumor protein (TCTP) is a multifunctional protein that plays a wide variety of physiological and pathological roles, including as a cytoplasmic repressor of Na,K-ATPase, an enzyme pivotal in maintaining Na^+^ and K^+^ ion gradients across the plasma membrane, by binding to and inhibiting Na,K-ATPase. Studies with transgenic mice overexpressing TCTP (TCTP-TG) revealed the pathophysiological significance of TCTP in the development of systemic arterial hypertension. Overexpression of TCTP and inhibition of Na,K-ATPase result in the elevation of cytoplasmic Ca^2+^ levels, which increases the vascular contractility in the mice, leading to hypertension. Furthermore, studies using an animal model constructed by multiple mating of TCTP-TG with apolipoprotein E knockout mice (ApoE KO) indicated that TCTP-induced hypertension facilitates the severity of atherosclerotic lesions in vivo. This review attempts to discuss the mechanisms underlying TCTP-induced hypertension and related diseases gleaned from studies using genetically altered animal models and the potential of TCTP as a target in the therapy of hypertension-related pathological conditions.

## 1. Introduction

Arterial hypertension, a major cause of premature death, remains the most prevalent risk factor that is implicated in the development of cardiovascular disease (CVD) and its complications [1]. Globally, the prevalence of hypertension in adults is nearly 32% and 34%, in women and men, respectively [2]. Uncontrolled or untreated high blood pressure (BP) over an extended period of time leads to damage to target organs, including the heart, kidneys, eyes, brain, and arterial blood vessels, causing functional and structural alterations, which could be the clinical manifestations of hypertension-related complications, such as atherosclerosis, heart failure, and chronic kidney disease (CKD) [3]. The dysfunction of vascular smooth muscle cells (VSMCs), among others, that shows the dysregulation in calcium signaling, migrative, and contractile properties is responsible for the mechanism underlying the development of arterial hypertension [3]. Therefore, understanding the molecular players and the modalities and modulation of their function has long been an area of extensive investigation in the field of pathophysiology of hypertension. 

Translational research using a variety of animal models that reflect the genetic and environmental factors has significantly contributed to the understanding of the pathophysiologic processes of hypertensive disorders. These efforts have led to the discovery of promising anti-hypertensive drug targets. Among those, Na,K-ATPase, also called the sodium pump, and its repressor, translationally controlled tumor protein (TCTP), have long been the theme of our research using TCTP-overexpressing and TCTP-deficient transgenic mouse models. Cumulative evidence has established that dysregulation of the sodium pump is involved in the development of hypertension and related complications, and that inhibitors of Na,K-ATPase can induce the hypertensive status. Prolonged suppression of Na,K-ATPase by ouabain, a well-established extracellular inhibitor of the sodium pump, has been shown to result in the development of hypertension in vivo [4,5,6]. Analogous to the action of ouabain, transgenic overexpression of TCTP, an intracellular inhibitor of Na,K-ATPase, showed phenotypes of systemic arterial hypertension in vivo, suggesting the causative role of TCTP as an inhibitor of Na,K-ATPase. 

This review attempts to describe the current status of our understanding of the role of TCTP in hypertensive disorders and related diseases, such as atherosclerosis and cataracts, as a consequence of its ability to suppress Na,K-ATPase, and to collate what we have learned from the studies using genetically engineered animal models. Therefore, studies that examined the role of TCTP in the pathophysiology of arterial hypertension and potentially related conditions such as atherosclerosis, heart failure, and obesity, if any, were included for the description in the present review. In addition, studies that use genetically engineered mouse models whose TCTP expression was overexpressed or reduced either systemically or tissue-specifically, were included, whereas research that dealt with non-genetic models or genetic models unrelated to TCTP were excluded in this study.

## 2. Pathophysiological Consequences of Na,K-ATPase Inhibition

Na,K-ATPase is an essential cell surface enzyme, which maintains ion gradients between the intracellular space and extracellular fluids [7]. This pump consists of three distinctive subunits, which appear in several isoforms. These isoforms include four α isoform (α1~4) and three β isoform (β1~3) subunits and an optional third γ subunit, a member of a Phe-X-Tyr-Asp (FXYD) motif-containing FXYD family [8]. The catalytic α and β subunits are required for ion-pumping activity while the β subunit affects the activity of this pump. Members of the FXYD proteins regulate the Na,K-ATPase activity in specific tissues [8]. The catalytic α subunit that consists of three cytoplasmic domains and ten transmembrane (TM) domains offers the binding sites for Na^+^, K^+^, and ATP as well as for cardiotonic steroids (CTSs), such as ouabain [9]. The high degree of conservation of the ouabain-binding site of this enzyme implies the existence of potential endogenous regulators [8].

Several endogenous “digitalis-like” or “ouabain-like” factors have been identified as a circulatory moiety that inhibits Na,K-ATPase [7]. The identification of endogenous CTS-like activity and the observation that Na,K-ATPase subunits exist in several isoforms imply the hormone-like functions of CTS and receptor-like function of the sodium pump [10]. Numerous investigations have found that CTS in relation with the sodium pump mediates various signaling cascades related to cell growth and division, release of endothelin-1 (ET-1) by endothelial cells, and other effects in cells [10]. Thus far, two distinctive, but coupled, pathways have been identified in the cellular signaling of Na,K-ATPase not only as an ion pump but also as a signaling transducer [10,11,12,13]. Therefore, dysregulation of this pump in both pathways has been an area of attention due to its pathophysiologic association with hypertension and the complications thereof [9].

### 2.1. Inhibition of Pump Activity of Na,K-ATPase by CTS

Na,K-ATPase is a key player in osmotic equilibrium that transports Na^+^ in exchange for K^+^ by hydrolyzing ATP to establish an electrochemical gradient across the cell membrane. Electrochemical gradients can drive certain types of cotransporters or exchangers, such as the Na^+^/Ca^2+^ exchanger (NCX) [8]. In addition, ionic gradients are also indispensable for the regulation of membrane potential, electrical excitability, cell volume, renal reabsorption of ions, and nutrient transport [14]. Therefore, it has been reported that mutations of the sodium pump α1 are related to the development of secondary hypertension, endocrine syndrome, peripheral neuropathy, and neuromuscular disorders [15]. In addition, dysfunction of Na,K-ATPase substantially affects the osmotic equilibrium and several other essential functions that relate to the development of hypertension, cardiac hypertrophy, cataracts, diabetes, and other disorders [16,17,18]. 

Moreover, the extracellular domain of the sodium pump serves as a receptor for cardiac glycosides, such as ouabain, and binding of cardiac glycoside to the sodium pump induces inhibition of its enzymatic activity. In smooth muscle cells, astrocytes, and hippocampal neurons of rodents, CTSs such as ouabain inhibit α2- or α3-isoforms of Na,K-ATPase of the cell membrane that is in close proximity to the sarcoplasmic or endoplasmic reticulum (S/ER), and induces a transient elevation of sub-plasmalemma sodium concentration [10]. With the coupling of the Na,K-ATPase to that of NCX, the latter works in its reverse mode, which results in the local increase in cytosolic Ca^2+^ levels. This triggers Ca^2+^ release from the S/ER to the cytoplasm, which in turn initiates Ca^2+^-mediated intracellular signaling cascades [10]. The subtle elevation of cytosolic calcium may amplify the impact on intracellular Ca^2+^ stores, which induces considerable changes in the contractility of myocardial and smooth muscles [9]. The resultant arteriolar contraction is one of the mechanisms for essential hypertension [10,19]. 

Additionally, enhanced intracellular Ca^2+^ transients upon stimuli render vasomotor neurons and endothelial cells hyper-responsive. The synergistic action of neurotransmitter release by sympathetic neurons and elevated vascular reactivity increases arterial tone and peripheral vascular resistance, all of which are the traits of essential hypertension [19]. The involvement of the sodium pump in hypertension is corroborated by the fact that the inhibited expression of the smooth muscle-specific α2 subunit elevates basal blood pressure with enhanced sensitivity to angiotensin II (Ang II), whereas overexpression of this subunit reduces the basal BP and its sensitivity to Ang II [4]. In addition, abnormal elevation of sodium in the lens is related to the opacification of the lens cortex in age-related cataracts, and the alteration of Na,K-ATPase activity appears relevant to cataract formation in human and animal models [20]. 

### 2.2. Signal Transduction of Na,K-ATPase by CTS

It is noteworthy that low-dose ouabain treatment activates the specific signaling without affecting the intracellular ion concentrations [21]. Mounting evidence has shown that Na,K-ATPase is engaged in the cardiac glycoside-mediated signaling cascades [22]. In the context of the signal-transducing functions of Na,K-ATPase, this pump is capable of communicating with signaling molecules by ligand-induced conformational changes within the signalosome [10]. Ouabain binding to the Na,K-ATPase in caveolae activates not only inositol triphosphate receptor (IP_3_R)-mediated Ca^2+^ release from S/ER but also Src kinase-mediated signal cascades [8]. CTS-induced conformational changes occurring within the caveolar sodium pump α1 subunit, which is a negative regulator of Src, allow the activation of Src [9]. 

Currently, Na,K-ATPase complex with Src is regarded as a unique receptor for CTS [9]. Activated Src triggers the execution of downstream signaling cascades, such as epidermal growth factor receptor (EGFR) transactivation-mediated Ras/Raf/mitogen-activated protein kinase (MAPK) kinase (MEK)/extracellular signal-regulated kinase (ERK) 1/2 pathways [23], inducing gene activation [10]. Src is known to be a key player in the Na,K-ATPase-mediated inter-receptor crosstalk with EGFR [24]. Activated as well are phosphoinositide 3-kinase (PI3K) [25] and EGFR-transactivated generation of reactive oxygen species (ROS), and the latter explains the clinical role of Na,K-ATPase as an oxidant amplifier contributing to the progression of cardiac fibrosis [9]. In addition, Src activates the phospholipase C (PLC) pathway [26] that also contributes to the IP_3_-induced Ca^2+^ release [10]. The signaling pathways of the sodium transporter are assumed to be implicated in hypertension, renal disease, diabetes, metabolic disease, and cardiovascular diseases. When there are alterations in the signaling pathways, signaling receptor function would change, causing hypertension, cardiac hypertrophy, tissue fibrosis, or cancer [9]. 

### 2.3. Ouabain-Induced Hypertension and Related Diseases

Ouabain, a well-known positive ionotropic agent in the heart, is also an endogenous adrenal cortical hormone because it can be synthesized in the adrenal glands [27]. It is known that the infusion of ouabain for a prolonged period in rodents [4,5,6], elevation of endogenous ouabain [28], and ouabain-like factors [29] in the circulation are associated with the development of hypertension. The elevation of endogenous ouabain also facilitates the proliferation of myocardial and smooth muscle cells [27]. Circulating endogenous ouabain inhibits the activity of the ouabain-sensitive Na,K-ATPase α2 subunit [4] and inhibition of the sodium pump by cardiac glycosides, such as ouabain, is associated with the pathogenesis of hypertension [17], cataract [18], and diabetic diseases [16], through the elevation of the intracellular Ca^2+^ concentrations. 

## 3. TCTP, as an Intracellular Na,K-ATPase Suppressor

TCTP, alternatively referred to as fortilin, p23, and histamine-releasing factor (HRF), is a highly conserved multifunctional 172-amino-acid protein that is under a high degree of regulation both at transcriptional and translational levels [30]. TCTP plays fundamental roles in the regulation of cell cycle progression, apoptosis, autophagy, survival, stress responses, growth and development, and cytokine-like activities, among others [30]. Given the multiplicity of roles it plays, dysregulation of TCTP can lead to cardiovascular and metabolic diseases including systemic and pulmonary arterial hypertension [31], atherosclerosis, diabetes, carcinogenesis, and allergic and inflammatory disorders [32].

Our research group has identified TCTP as a cytoplasmic repressor for Na,K-ATPase [33]. In contrast to ouabain that binds to the outer part of the sodium pump, TCTP interacts with the cytosolic domain of Na,K-ATPase [33]. Searching for the cytoplasmic molecules that are potentially associated with the Na,K-ATPase α subunit, we screened a rat skeletal muscle DNA library using a yeast two-hybrid system. We found that the third large cytoplasmic domain (CD3) of Na,K-ATPase α1 and α2 isoforms interacts with the rat TCTP [33]. 

While studying the interaction of TCTP with the sodium pump in HeLa cells, we discovered the inhibitory effect of TCTP on Na,K-ATPase activity. Interaction of rat TCTP with the sodium pump does not affect the mRNA and protein levels of the Na,K-ATPase α subunit, indicating a relatively short-term regulation by TCTP [33]. TCTP inhibited the sodium pump activity in a dose-dependent manner [33]. In addition, the C-terminal region (102–172 residues) of rat TCTP was the moiety involved in the inhibition of pump activity of Na,K-ATPase [33]. These observations suggest that TCTP is possibly playing a promotive role in hypertension and raise the question of whether TCTP elevates cytosolic Ca^2+^ in vivo [33]. Interaction of the sodium pump with TCTP is now generally accepted to have important roles not only in the biological functions and signaling but also in the pathophysiology of human diseases [34,35,36].

It has been shown that TCTP plays either protective or causative roles in certain diseases in some conditions, but it is clear that the dysregulation of TCTP contributes to the development of various pathophysiological processes [37], as shown in several genetically altered mouse models (Table 1). Conventional or tissue/cell-specific TCTP-overexpressing transgenic mice (TCTP-TG) were used to study the gain-of-function of TCTP. Conversely, heterozygous TCTP-deficient mice (TCTP^+/−^), in which TCTP gene is deficient causing a loss of function, were studied because of embryonic lethality of homozygous TCTP knockout mice (TCTP^−/−^) [38]. This review focuses on the roles of TCTP in hypertension and related diseases, including atherosclerosis, cataracts, and other metabolic disorders that have been studied in genetically engineered animal models.

## 4. Pathophysiology of Hypertension in TCTP-TG

Animal models have been used for conducting etiological studies of genetic hypertension. These include spontaneous hypertensive rats (SHRs), Dahl salt-sensitive rats, and transgenic models generated by overexpression of particular genes [53]. To address the questions raised regarding the pathophysiological consequences of TCTP-mediated Na,K-ATPase inhibition in vivo, we constructed transgenic mice overexpressing TCTP (TCTP-TG). It was established in either C57BL/6×CBA hybrid or C57BL/6N inbred backgrounds using pCAGGS-TCTP cDNA construct consisting of cytomegalovirus enhancer (CMV-IE) and chicken β-actin promotor (Figure 1A). 

Their phenotypic characteristics are partly affected by the genetic background of TCTP-TG because, for example, cataractogenesis was profound in C57BL/6N background mice [41] while both types of transgenic mice show hypertensive properties regardless of their genetic backgrounds [39,41,44]. Because of the decreased fertility of inbred offspring, TCTP-TG was generated in C57BL/6×CBA hybrid strains in the majority of our investigations. For the study regarding atherogenesis, TCTP-TG was subjected to crossbreeding to generate apolipoprotein (ApoE)-deficient TCTP-TG (ApoE-KO/TCTP-TG) with a hybrid background (Figure 1B). 

### 4.1. Systemic Arterial Hypertension by Na,K-ATPase Inhibition in TCTP-TG

Both C57BL/6×CBA and C57BL/6N mice overexpressing TCTP showed phenotypes of hypertension and hybrid mice were used in the study. The gender-related dimorphism regarding BP was observed, also reported in heterozygous female TGR(mRen2)27 transgenic rats [54], possibly due to the differences in hormonal changes or vascular responsiveness between female and male TCTP-TG mice occurring with ageing [39]. At approximately 6 weeks after birth, systemic arterial hypertension developed in both male and female TCTP-TG but after 19–20 weeks female TCTP-TG did not exhibit a significant difference in BP compared with non-transgenic (nTG) mice [39]. Therefore, heterozygous male TCTP-TG mice were studied for further investigations. 

As with arterial BP elevation, an increase in left ventricular systolic pressure (LVSP) was found in TCTP-TG mice [39]. The thickening of the left ventricular posterior walls was observed in 15- to 16-month-old male TG mice, possibly caused by the sustained elevation of BP, but TCTP-TG did not exhibit the significant cardiac dysfunction [39]. TCTP-TG showed the characteristics of hypertension: first, the alteration of vascular functions that exhibit enhanced contractility of aortic rings devoid of endothelial layer, and secondly, augmented sensitivity to vasoconstrictor stimuli such as norepinephrine (NE) and serotonin (5-HT). Moreover, vascular relaxation reactivity following sodium nitroprusside (SNP) treatment after NE-induced contraction was decreased, indicating the reduced sensitivity to vasodilators [39]. Impaired vasodilatory responses accompanied by the hypercontractile profile in the vasculature isolated from TCTP-TG mice aberrantly elevated the BP [39]. 

As expected, vascular smooth muscle cells (VSMCs) isolated from TCTP-TG showed suppressed activity of Na,K-ATPase α1 and α2 isoforms that are the predominant Na,K-ATPase in mouse aorta, confirming the TCTP-induced sodium pump inhibition in VSMCs [39]. The expression of α1 in VSMCs from TCTP-TG (TG-VSMCs) was elevated, whereas that of α2 did not show any significant changes [39]. Because transient TCTP overexpression in HeLa cells did not affect the Na,K-ATPase expression, as observed in our earlier studies [33], elevated α1 expression in TCTP-TG mice suggests a potential compensatory mechanism due to the prolonged pump inhibition in vivo [39]. 

Then, we confirmed that inhibition of Na,K-ATPase activity in TG-VSMCs induces augmented intracellular Ca^2+^ mobilization, possibly through the reverse mode of NCX. Upon 5-HT stimuli, TG-VSMCs exhibited elevated Ca^2+^ transients that were mobilized from the ER [39]. Of interest, ouabain treatment at a concentration that blocks almost all the activity of Na,K-ATPase did not further enhance the 5-HT-evoked Ca^2+^ transients in TG-VSMCs whereas that of non-transgenic mice was elevated by ouabain, compared with untreated cells [39]. This phenomenon can be explained by the fact that ouabain may have negligible impact on TCTP-TG whose homeostasis has already been established by TCTP-induced Na,K-ATPase inhibition [39]. Likewise, VSMCs of TCTP-TG showed increased sensitivity to vasoconstrictor, elevated levels of Ca^2+^ in the resting state, and increased capacity for intracellular Ca^2+^ mobilization, and ultimately enhanced contraction by stimuli, accompanied by reduction in vasorelaxant-induced relaxation [39]. 

Taking these observations together, we delineated the pathophysiology of TCTP-induced hypertension using TCTP-TG, as depicted in Figure 2. Repression of Na,K-ATPase by TCTP overexpression results in the rise of cytosolic Na^+^ level, and subsequent elevation of cytosolic Ca^2+^ mobilization and Ca^2+^ storage, possibly through the reversion action of NCX, ultimately inducing the hyper-contractility of VSMC in TCTP-TG. Clearly, based on the fact that high blood pressure implies the dysfunctions in cardiac and/or arterial parameters [55], non-significant cardiac malfunctions indicate that TCTP-TG-induced hypertension is mediated by vascular malfunctions, as shown by elevated contractility and reduced relaxation in VSMC from TCTP-TG mice. This phenomenon is similar to that of ouabain treatment-induced hypertension exhibiting augmented vascular reactivity to vasopressors in renal artery of rats [56], confirming that TCTP can be viewed as an endogenous ouabain-like entity. Based on the observation that TCTP^+/−^ and wild-type mice did not show a significant difference in systolic BP [45], overexpression of TCTP might be essential in inducing arterial systemic hypertension in vivo. 

### 4.2. Upregulation of RhoA/Rho Kinase Signaling in TCTP-TG

The contractility of smooth muscle cells is largely determined by the phosphorylation status of the regulatory myosin light chain (MLC) whose activity is under regulation by myosin light chain kinase (MLCK) and myosin light chain phosphatase (MLCP) [57]. Vasoconstrictors like Ang II binding to G-protein coupled receptors (GPCR) activates G_q/11_, which in turn activates phospholipase C (PLC)-β and elevates Ca^2+^ concentration, thereby stimulating MLCK activity and induces G_12/13_ to activate RhoA/Rho kinase and, in turn, inhibits MLCP. A myosin binding subunit, the myosin phosphatase target unit (MYPT-1) of MLCP, is one of the most important substrates of Rho kinase [57]. Both pathways together converge into the MLC phosphorylation and enhance the contraction of VSMCs and myofibrils [57]. As a downstream mediator of Ang II signaling, a small GTP-binding protein, RhoA, which maintains MLC phosphorylation in VSMCs, regulates Ca^2+^ sensitivity of the myofilaments through MLCP inhibition [58,59]. 

It is generally accepted that Ca^2+^ sensitivity of myofilaments is under regulation by RhoA/Rho kinase-induced MLCP inhibition [58,59] and that dysregulation of RhoA/Rho kinase signaling results in the hypercontractile properties of VSMC, thereby contributing to the pathogenesis of hypertension [57]. Upregulation of RhoA/Rho kinase signaling pathway has been described in cardiovascular diseases, such as systemic and pulmonary hypertension, atherosclerosis, heart failure, and stroke [57,60] and it is deeply involved in cardiovascular and renal pathophysiology [61]. Consistently, dysregulation of RhoA/Rho kinase pathway has been reported in several hypertensive rat models, including SHRs, renal hypertensive rats, and deoxycorticosterone acetate (DOCA) salt-induced hypertensive rats [58].

We found that TCTP-induced RhoA/Rho kinase upregulation is involved in the hypertensive phenotype of TCTP-TG [40]. Increased contraction in the aorta from TCTP-TG under lower concentrations of K^+^ [39] implies the potential dysregulation of pathways regulating Ca^2+^ sensitization, such as RhoA/Rho kinase pathway. When we examined these alterations in aortas isolated from TCTP-TG mice, elevated RhoA expression and phosphorylated MLC (p-MLC) were confirmed, whereas aortas obtained from heterozygous mice containing a deleted allele of TCTP (TCTP^+/−^) showed a decrease in RhoA expression and p-MLC in their aortas [40].

Additionally, overexpression of TCTP increased RhoA expression and activated its downstream signaling, including MYPT-1 and MLC phosphorylation in primary cultured VSMC. In contrast, knockdown of TCTP expression in VSMCs showed suppression of RhoA expression as well as Rho kinase signaling pathways [40]. Inhibition of Rho kinase signaling can be an alternative approach in the modulation of TCTP-induced hypertension. It is evident that dysregulation of RhoA-related Ca^2+^ sensitization mechanism partly underlies the hypertensive phenotypes of TCTP-TG (Figure 2). 

### 4.3. Elevation of the Expression of Peroxiredoxin 3 (Prx3) and Heat Shock Protein 25 (Hsp25) in TCTP-TG

A proteomic analysis to delineate the potential pathophysiological changes in TCTP-TG mice of C57BL/6×CBA strain showed differentially expressed proteins involving reactive oxygen species (ROS) metabolism, fatty acids/amino acids metabolism, energy metabolism, and cytoskeletal organization in heart tissue of TCTP-TG compared with those of control mice [62]. Of note, the expression of a mitochondrial antioxidant enzyme, peroxiredoxin 3 (Prx3) showed decrease in 9-week-old TCTP-TG mice, whereas cytosolic Prx2 expression was elevated in 19-week-old TCTP-TG [62]. Because mitochondrial oxidative stress takes part in the pathogenesis of hypertension [63,64,65], altered expression of these enzymes implies the potential relation of ROS in TCTP-induced hypertension [62]. The protective and anti-inflammatory roles of Prx3 in various cell types were suggested [66] and Prx3 overexpression protects the heart from the ventricular remodeling and heart failure following myocardial infarction by attenuating mitochondrial oxidative stress [67]. Additionally, the higher Hsp25 expression in 9-week-old TCTP-TG mice, also indicates the possible role of ROS in TCTP-induced hypertension. Stress-inducible Hsp25 is upregulated in the kidney isolated from the rats with prolonged Ang II administration, and its expression was reported to be involved in the regulation of BP [68]. Studies on the potential roles of Prx3 and Hsp25 in the heart of TCTP-TG mice are an area of interest for the further studies.

## 5. Pathophysiology of Cataractogenesis in TCTP-TG

There is a 2.13 times greater risk of having a cataract extraction in hypertensive patients, especially in the 60–69 year age group, compared with non-hypertensives individuals [69]. A meta-analysis also indicates that hypertension enhances the risk of cataracts [70]. In salt-sensitive hypertensive Dahl rats, the incidence of cataracts was 30–35% and it can be predicted by the suppression of Na,K-ATPase, suggesting a defect in this pump in hypertensive rats [71,72,73].

Cataractogenesis, an impaired lens transparency, results from the dysregulation of ionic and fluid homeostasis [41], which is tightly regulated by Na,K-ATPase in the lens cells [20,74]. Elevated sodium and calcium concentrations or digitalis-like compounds have been found in human cataractous lenses [75]. Sodium pump inhibitors like ouabain induce the elevation of the sodium level with a concomitant decrease in the potassium level in the lens, and such osmotic imbalance leads to the accumulation of fluids in the lens, and then cells swell and degrade, resulting in the formation of fluid droplets, the cause to opacify [76,77,78]. In addition to fluid-induced cell death, elevated sodium levels due to the Na,K-ATPase repression induce the rise in cytosolic level and transient of Ca^2+^, which activates Ca^2+^-dependent proteolytic enzymes, cellular integrity impairment and apoptosis, eventually resulting in the loss of lens transparency [76,77,78].

Of interest, the elevation of BP is not necessarily the only effect of TCTP-induced Na,K-ATPase inhibition in the TCTP-TG because an elevated incidence rate of cataracts was found in TCTP-TG mice of C57BL/6N inbred strain, compared to that of non-TG (7.38% vs. 1.47%) at 10 weeks after birth [41]. Similar to the majority types of human cataracts, TCTP-TG showed mixed types of nuclear, cortical, and anterior subcapsular opacification [41]. For still unknown reasons, this phenotype was not prominent in C57BL6×CBA hybrid TCTP-TG although both strains of TCTP-TG elicited hypertension as a pathophysiological consequence of Na,K-ATPase inhibition [41]. This mode of cataractogenesis operates in human lens epithelial cells because TCTP overexpression induced an increase in cytosolic Ca^2+^ mobilization upon histamine stimuli via sodium pump inhibition, which might induce the proteolytic activity of enzymes (Figure 2). The transient overexpression of TCTP in lens epithelial cells did not affect the expression of the Na,K-ATPase α-subunit [41]. 

Interestingly, TCTP appears to affect the normal eye development because some TCTP-TG mice showed abnormal eye development, including unusual sizing of the eyes, and even absence of the eye [41]. This intriguing observation emphasizes the need to identify the fundamental roles of TCTP in the developmental processes of the eyes. 

## 6. Pathophysiology of Atherosclerosis in TCTP-TG and TCTP^+/−^ Mice

### 6.1. Exacerbation of Atherosclerosis by TCTP-Induced Hypertension in TCTP-TG/ApoE KO

Atherosclerosis is a pathological remodeling of arterial walls with the development of atheromatous plaques involving endothelial dysfunction, migration of SMCs, and accumulation of cholesterol in the lesions [79,80]. Arterial hypertension is a key etiological factor in the development of atherosclerosis and, thus, of cardiovascular disorders [79], through the various pathological links including hemodynamic stress [81]. Ang II- and high salt-induced hypertension facilitated the development of atherogenesis in apolipoprotein E-deficient mice (ApoE KO) [82,83] whereas Ang II-induced atherosclerosis was attenuated by enalapril treatment in vivo because of its anti-atherogenic and anti-inflammatory effects [84]. In clinical settings, angiotensin-converting enzyme inhibitors (ACEIs) and angiotensin receptor blockers (ARBs) retard the pathogenic process of atherosclerosis in hypertensive patients [85]. 

Genetically modified animal models of atherosclerosis include apolipoprotein E knockout mice (ApoE KO), low-density lipoprotein (LDL) receptor knockout mice (LDLR^−/−^), scavenger receptor class B member 1 knockout mice (SR-BI KO), *db/db* and *ob/ob* mice, Zucker fatty rats, and cholesterol ester transfer protein (CETP) transgenic rats [53]. Among those, the most extensively investigated is the ApoE KO that lacks ApoE, a glycoprotein ligand for receptors that eliminates chylomicron remnants and very low-density lipoproteins (VLDLs), which exhibits spontaneous hypercholesterolemia and the formation of atherosclerotic lesions in vessels [86,87].

Our group investigated whether TCTP overexpression accelerates the process of atherogenesis in animal models [44]. Multiple crossbreeding of TCTP-TG with ApoE-null mice generated nTG/ApoE KO and TCTP-TG/ApoE KO mice having comparable genetic backgrounds (about 97% C57BL/6 and 3% CBA) (Figure 1B). TCTP-TG and nTG mice carrying identical genetic makeup were generated by backcrossing the TCTP-TG C57BL/6×CBA hybrid mice to C57BL/6 mice. The differences in body weight changes and major organ weights were insignificant among the four groups, but the brain, lung, and spleen weights showed an increase in TCTP-TG/ApoE KO compared to TCTP-TG [44]. Following the lipid-enriched western diet of 7-week-old mice for 16 weeks, lipid-laden atherosclerotic plaques were found in the thoracic aorta of ApoE-null mice, but not in that of TCTP-TG [44]. Importantly, the transgenic overexpression of TCTP in ApoE KO showed an exacerbation of atherosclerotic lesions, exhibiting a wider distribution and thicker lesions than that of ApoE KO [44]. The BP elevation of TCTP-TG and TCTP-TG/ApoE KO was comparable [44]. Based on the observation that TCTP-TG shows negligible atherosclerotic lesion formation, hypertension alone was not enough to establish atherosclerotic lesions. Moreover, high-fat diet feeding in TCTP-TG also did not cause atherosclerosis and the animals exhibited normal lipid metabolism [44], suggesting that ApoE genetic malfunction may be a prerequisite for establishing atherosclerosis.

We found that TCTP overexpression in an ApoE KO genetic background did not affect the plasma glucose, albumin, and lipid profiles of ApoE KO mice [44], indicating that atherosclerotic acceleration by TCTP is not mediated by those factors, including increased burden of plasma lipid. Therefore, we speculated that hemodynamic stress or inflammatory responses provoked by high BP [81,88] might be the proatherogenic factors that operate in TCTP-induced atherogenic progression [44]. The potential involvement of ET-1, a potent endothelium-derived vasoactive peptide that is upregulated in atherogenic status to facilitate the proliferation, migration, and contraction of VSMC [89,90,91], was also proposed. Such a hypothesis can be supported by the findings that Na,K-ATPase inhibition, such as that which results from ouabain treatment, induces ET-1 release from the endothelial cells [92,93,94,95,96] and that TCTP synthesis was upregulated following stimulation of human lung fibroblasts with ET-1 [97], though the interplay between TCTP and ET-1 and its consequences require experimental verification.

Our unpublished data on ligation-induced atherosclerotic mouse model also supports the role of TCTP in the atherogenesis. Two weeks after the ligation operation on the left common carotid artery (LCCA), ligated carotid arteries of TCTP-TG showed a 3.4-fold increase in intimal area compared to that of non-TG, suggesting TCTP-induced exacerbation of atherosclerosis through intimal layer thickening.

### 6.2. Aggravation of Atherosclerosis by TCTP-Induced Survival of Macrophages in TCTP^+/−^ Mice

Pinkaew et al. described the pathophysiological roles of TCTP in protecting macrophages in atherosclerosis using heterozygous TCTP-deficient mice [45]. This group noted that TCTP, not hypertension per se, can facilitate atherosclerosis by protecting macrophages against apoptosis. They constructed mouse models of TCTP-deficient (TCTP^+/−^) and wild-type (TCTP^+/+^) mice on a hypercholesterolemic genetic background, Ldlr^−/−^ Apobec1^−/−^ whose low-density lipoprotein receptor (*Ldlr*) and apolipoprotein B mRNA editing enzyme, catalytic polypeptide 1 (*Apobec 1*) genes are lacking. Ldlr^−/−^ Apobec1^−/−^ mice have considerably elevated blood cholesterol levels on a normal diet and induce atherogenic processes from fatty streaks through the fibrous cap [45]. Of note, this group tried to find the role of TCTP itself by ruling out the effect of BP as there were no significant differences in BP between TCTP^+/−^ and TCTP^+/+^ [45]. 

On a normal diet for 10 months, both groups of mice showed comparable BP and lipid profiles, but TCTP^+/−^ had significantly less atherosclerosis in aortas compared to TCTP^+/+^ littermate controls [45], indicating that TCTP deficiency is related to a lesser degree of atherosclerosis. It was found that TCTP^+/−^ mice had fewer macrophages within atherosclerotic plaques and more apoptotic macrophages in the intima than those of TCTP^+/+^ mice [45]. Moreover, peritoneal macrophages isolated from TCTP^+/−^ mice showed elevated expression of Bax as well as apoptosis in the plaques both at baseline and upon oxidized LDL stimuli [45]. TCTP expression in peritoneal macrophages was more enhanced in hypercholesterolemic sera from Ldlr^−/−^ Apobec1^−/−^ mice than that from control mice. Additionally, macrophage colony-stimulating factor (M-CSF), which is elevated in the plasma of patients with coronary artery disease [98], induced TCTP expression [45]. Based on these findings, they concluded that the pro-atherosclerotic microenvironment, including hypercholesterolemia and inflammatory cytokines like M-CSF, allows the induction of TCTP expression in macrophages and this protects macrophages against Bax-induced apoptosis, thereby facilitating the propagation of macrophages and atherosclerosis [45]. Moreover, TCTP levels are abundantly elevated in human atherosclerotic plaques, showing a positive correlation with the severity of lesions from the fatty streak to the fibrous plaques [45]. 

A recent study using patients’ data concluded that elevated TCTP in atherosclerotic lesions might result from the adaptive responses to profound apoptosis [99]. Higher plasma TCTP levels were found in the patients with coronary artery disease (CAD), particularly those with three-vessel disease (3VD) [99]. Furthermore, the degree of TCTP levels is positively correlated with the severity of CAD, suggesting it as a biomarker for CAD [99]. It was speculated that insufficient clearance of extensive apoptotic cells in atherosclerotic lesions might cause the accumulation of apoptotic cells and inflammatory responses that in turn induce the adaptive responses involving TCTP expression to reduce the excessive apoptosis [99]. 

Several lines of evidence unequivocally show the facilitative role of TCTP overexpression in atherosclerotic plaque formation via TCTP-induced hypertension or by reducing apoptosis in macrophages [44,45]. In this review, TCTP serves as a novel target for the modulation of hypertension-related atherosclerosis.

## 7. Role of TCTP in Hypertension-Related Diseases

Studies using genetically altered animal models, including conventional or tissue-specific TCTP-overexpressing and TCTP-deficient mice, have shed light on TCTP’s potential pathophysiological role in chronic diseases, such as osteoporosis [42], rheumatoid arthritis [43], and allergy [46]. Other studies have indicated the protective roles of TCTP due to its anti-apoptotic and cell-protective functions. Such beneficial effects were found in the studies regarding diabetes [48], heart failure [49,50], and liver damage [51,52]. Additionally, TCTP promotes the energy expenditure and metabolic homeostasis that might be preventive of obesity-related metabolic disorders [47] (Table 1). Here, we briefly discuss the protective roles of TCTP in some conditions, including obesity and heart failure, among others, to help understand the plethora of TCTP functions in cardiovascular and metabolic diseases in vivo.

### 7.1. Obesity

Obesity, characterized by excessive accumulation of body fats, is a representative of the modifiable risk factors for cardiovascular diseases including hypertension and atherosclerosis [100]. Based on the fortuitous finding that TCTP-TG mice with C57BL/6N background have the phenotypes of relatively lighter body weight than control mice, our group investigated the impact of TCTP on metabolic tissues and systemic energy metabolism [47]. We confirmed that TCTP-TG mice of C57BL/6N strain under normal chaw diet (NCD) showed significantly lower body weight compared to wild-type mice (WT) as early as six weeks after birth despite their comparable amount of food intake [47]. TCTP-TG showed an increase in muscle weight and reduction in fat mass due to inhibition of the hypertrophy of adipocytes in epididymal white adipose tissue (WAT). In addition, TCTP-TG exhibited improvement in hepatic lipid accumulation, plasma lipid profiles, and glucose tolerance as well as overall energy expenditures [47]. When we compared the metabolic homeostasis between TCTP-TG and WT under NCD and high-fat diet (HFD) conditions, TCTP-TG indicated improved metabolic homeostasis under both conditions, with the enhanced glucose tolerance and insulin sensitivity [47]. 

Upon cold exposure that triggers thermogenesis by the sympathetic nerve system (SNS) activation in the brown adipose tissue (BAT), TCTP-TG showed improved adaptive thermogenesis [47]. TCTP overexpression attenuated systemic metabolic imbalance by upregulating the uncoupling protein 1 (UCP1)-mediated thermogenesis in the BAT, serving as a modulator in the process of energy expenditure [47]. Affluent mitochondria contained in the BAT, where mitochondrial carrier protein, UCP1, mediates thermogenesis by sympathetic stimuli and produces heat by consuming triglycerides [101]. BAT activation is reported to reduce hypercholesterolemia and to exert protective roles from atherogenesis [102]. More importantly, TCTP-TG exhibited resistance to HFD-induced obesity and metabolic disorders [47]. The increase in energy expenditure and thermogenesis of BAT under HFD in TCTP-TG underscores the critical roles of TCTP in metabolic homeostasis through energy expenditure [47]. Therefore, TCTP can be viewed as a rational target for the energy expenditure-related conditions such as obesity and metabolic disorders [47].

### 7.2. Heart Failure

A recent publication by Cai et al. addressed TCTP’s role in cardiomyocyte survival based on the observation that an animal model overexpressing cardiomyocyte-specific TCTP-TG showed a protective role in heart failure [49]. Cardiomyocyte-specific TCTP overexpression drastically reduced the susceptibility to doxorubicin (DOX)-induced cardiac dysfunction in mice [49]. It also inhibited the induction of Bcl-2/adenovirus E1B 19 kDa-interacting protein 3 (Bnip3), a molecule that mediates the TCTP-loss-induced cardiomyocyte death [49]. In mice with cardiomyocyte-specific TCTP overexpression, treatment with dihydroartemisinin, a pharmacological TCTP inhibitor, did not induce heart failure and cardiomyocyte death, both of which were induced in control mice [49]. Altogether, these observations show that TCTP is essential for cardiomyocyte survival and can be suggested as a therapeutic target that ameliorates DOX-induced heart failure [49]. 

In addition, studies in an animal model that lacks TCTP in their heart confirm the protective role of TCTP in heart failure [50]. Mice devoid of TCTP expression in the heart showed premature death by 9 weeks of age because of extensive apoptotic cardiomyocytes and severe heart failure [50]. Heart-specific TCTP KO mice showed upregulation of p53 target genes in their hearts and heart-targeted deletion of p53 in those mice prolonged survival of mice from 9 to 18 weeks by preventing apoptosis of cardiomyocytes. Therefore, this group concluded that inappropriate expression of TCTP is related to the pathophysiology of heart failure [50].

## 8. Conclusions

New perspectives on the pathophysiological roles of TCTP as a multifunctional protein as well as a cytosolic sodium pump inhibitor and as a player in the development of systemic and arterial hypertension, cardiac hypertrophy, cataracts, diabetes, and related disorders via inhibition of the sodium pump in VSMC, lens epithelial, or other relevant cells have emerged from the studies of genetically engineered animal models. Additionally, TCTP accelerates the pathogenesis and severity of atherosclerosis by the mechanisms involving its anti-apoptotic activity on macrophages in atherosclerotic lesions. Conversely, modalities that modulate TCTP could be a promising strategy in the therapeutics for systemic hypertension, hypertension-induced atherosclerosis, and cataracts, all of which are associated with Na,K-ATPase suppression.

Intriguingly, TCTP also showed beneficial effects on heart failure and obesity by protecting the cells from apoptosis and enhancing the metabolic expenditure, respectively, in animal models. Because TCTP has a plethora of physiological functions and is essential for the survival and growth of cells and organisms, a certain extent of TCTP expression in vivo seems indispensable for maintaining normal physiology. However, not only excessive but also deficient expression of TCTP is related to the pathophysiology in specific conditions, as shown in the phenotypes of TCTP-TG and TCTP^+/−^ mice. Further studies regarding hitherto unknown potential mechanisms and roles of TCTP in terms of various cellular functions might delineate its pathophysiological network in the cardiovascular conditions. In this perspective, possible implications of TCTP in the cellular senescence and its contributory mechanism in the arterial hypertension and related disorders can be one of the future studies based on the causative roles of senescence in the majority of cardiovascular diseases [103,104]. 

It is challenging, at present, to describe the complicated networking between protective and causative roles of TCTP in a unified figure, and disease-specific mechanisms, and regulations affecting the expression and activity of TCTP need to be delineated for future investigations. Clearly, studies in disease-specific animal models and human cases may help us understand how TCTP exerts manifold functions in a certain disease and pave the way for TCTP-targeting therapeutics.

## Figures and Tables

**Figure 1 biomedicines-10-02722-f001:**
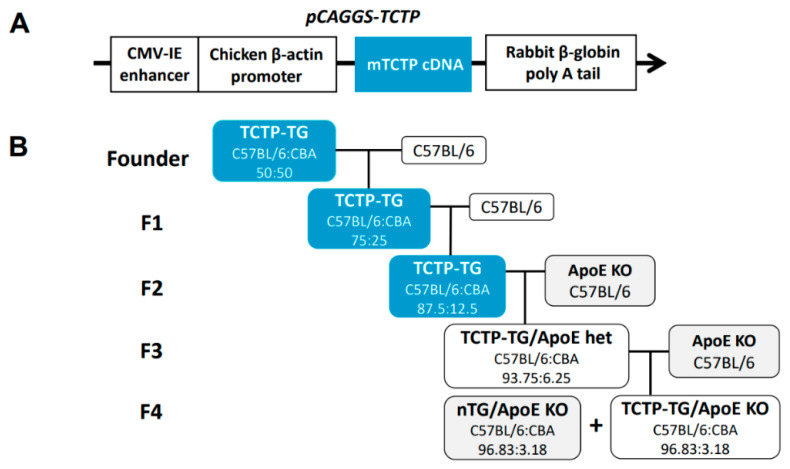
Generation of TCTP-TG and TCTP-TG/ApoE KO mice. (**A**) Schematic representation of the gene construct used for TCTP transgenic overexpression in mice. The TCTP transgene of pCAGGS-TCTP cDNA containing CMV-IE enhancer and chicken β-actin promoter constructed in the transgenic expression vector pCAGGS was used to generate the transgenic mice. (**B**) Strategy of crossbreeding for generation of nTG/ApoE KO and TCTP-TG/ApoE KO mice exhibiting identical genetic backgrounds.

**Figure 2 biomedicines-10-02722-f002:**
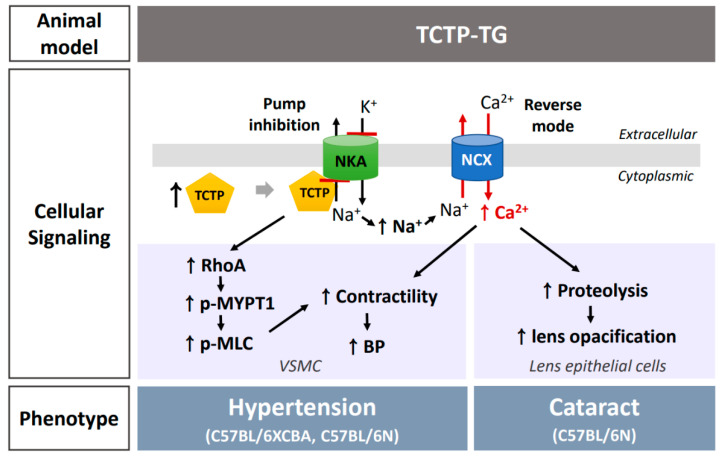
TCTP overexpression-induced consequences in transgenic TCTP-overexpressing (TCTP-TG) mice. Transgenic overexpression of TCTP led to the Na,K-ATPase inhibition, which results in the development of hypertension through elevated cytosolic Ca^2+^ levels and vascular contractility and tone. Such an increase in Ca^2+^ can also occur in the lens epithelial cells of TCTP-TG, at this time, inducing the activation of proteolytic enzyme and lens swelling, all of which affect the cataractogenic processes. RhoA/Rho kinase activation and resultant MYPT-1 phosphorylation in TCTP-TG appears to partly contribute to the increased sensitivity of myofibrils upon constrictor stimuli, thereby increasing the contractility of VSMC in TCTP-TG (↑ indicates increase).

**Table 1 biomedicines-10-02722-t001:** Genetically modified mouse models that exhibit elevated or reduced expressions of TCTP and its phenotypes and pathophysiological significances in vivo.

Animal Models	Related Disease	Phenotypes	Ref.
**Pathophysiological roles of TCTP**			
TCTP-TG	Hypertension	Development of systemic arterial hypertension and increased vascular contractility ↑ RhoA expression and phospho-myosin light chain (p-MLC) in aorta	[39,40]
	Cataract	↑ Incidence of cataract formation in lens ↑ Abnormal eye development	[41]
	Osteoporosis	↓ Bone mass in femurs ↑ Osteoporotic features in femur bones, and osteoclast cell count	[42]
	Rheumatoid arthritis	↑ Inflammatory responses, bone erosion, and cartilage destruction upon collagen-induced arthritis (CIA)	[43]
TCTP^+/−^	Hypertension	↓ RhoA expression and p-MLC in aorta	[40]
	Osteoporosis	↑ Bone mass in femurs ↓ Osteoporotic features in femur bones, and osteoclast cell count	[42]
	Rheumatoid arthritis	↓ Synovial inflammation, bone erosion, cartilage damage, and osteoclastic bone resorption upon CIA	[43]
TCTP-TG/ ApoE-KO	Atherosclerosis	↑ Exacerbation of atherosclerotic lesion formation by high-fat diet without alteration in plasma lipid profiles, compared with ApoE-KO mice	[44]
TCTP^+/−^ Ldlr^−/−^Apobec1^−/−^	Atherosclerosis	Similar lipid profiles and BP compared with TCTP^+/+^ Ldlr^−/−^ Apobec1^−/−^ mice ↓ Atherosclerotic lesion in aorta and macrophage numbers in lesions, compared to TCTP^+/+^ Ldlr^−/−^ Apobec1^−/−^ mice ↑ Bax expression and apoptosis of peritoneal macrophages in the intima layer, compared with TCTP^+/+^ Ldlr^−/−^ Apobec1^−/−^ mice	[45]
Clara cell-specific TCTP-TG	Allergic asthma	↑ Allergic and asthmatic inflammation with increase in serum and bronchoalveolar lavage (BAL) IgE, interleukin-4 (IL-4), and eosinophil count upon ovalbumin (OVA) challenge ↑ TCTP secretion and macrophage counts in BAL fluids	[46]
**Protective roles of TCTP**			
TCTP-TG	Obesity	↑ Metabolic homeostasis under both normal and high-fat diet conditions with enhanced glucose tolerance and insulin sensitivity ↑ Energy expenditure with upregulation of uncoupling protein 1 (UCP1) in the brown adipose tissue (BAT) ↑ Adaptive thermogenesis of BAT after cold exposure	[47]
Tregs-specific TCTP-TG	Diabetes	↑ Forkhead box protein P3(FOXP3) expression and prolonged survival of regulatory T cells (Tregs) ↓ Development of autoimmune diabetes by inhibiting the apoptosis of Tregs	[48]
Cardiomyocyte-specific TCTP-TG	Heart failure	↓ Doxorubicin-induced cardiac dysfunction and Bcl-2 interacting protein 3 (Bnip3) induction ↓ Dihydroartemisinin-induced heart failure and cardiomyocyte death	[49]
Heart-specific TCTP-KO	Heart failure	Mice lacking TCTP in the heart die by 9 weeks of age because of severe heart failure and extensive cardiomyocyte apoptosis	[50]
Liver-specific TCTP-TG	Liver damage	↑ Peroxiredoxin-1 (PRX1) activity in the liver and protection against alcohol, and ROS-mediated liver damage	[51]
Liver-specific TCTP-KO	Liver damage	↓ Endoplasmic reticulum (ER) stress-induced liver failure and death by blocking apoptosis in the liver	[52]

↑ indicates increase; ↓ indicates decrease.

## Data Availability

Not applicable.

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
