# Peer review of "Role of Translationally Controlled Tumor Protein (TCTP) in the Development of Hypertension and Related Diseases in Mouse Models"

_biomedicines, 2022, doi:10.3390/biomedicines10112722_

Round 1

Reviewer 1 Report

I revised the manuscript entitled "Role of Translationally Controlled Tumor Protein (TCTP) in 2 the Development of Hypertension and Related Diseases in 3 Mouse Models". This is an excellent piece of work. I will only suggest a brief comment in the discussion section for the role of TCTP in cellular senescence and its possible relation with the pathophysiology of arterial hypertension (see Evangelou et al Physiol Rev 2022, Hu C, et al. Aging Dis. 2022)

Author Response

Dear Reviewer 1:

Thank you for providing me with the comments from reviewers of our manuscript entitled “Role of Translationally Controlled Tumor Protein (TCTP) in the Development of Hypertension and Related Diseases in Mouse Models”. We are now electronically submitting a revised manuscript, prepared taking into account the helpful critiques of the reviewers. We are grateful for this input, which we believe has strengthened the manuscript. We have provided below our detailed responses to the specific comments of the reviewers. These responses have also been included in the manuscript.

Reviewer: 1

I revised the manuscript entitled "Role of Translationally Controlled Tumor Protein (TCTP) in the Development of Hypertension and Related Diseases in Mouse Models". This is an excellent piece of work.

I will only suggest a brief comment in the discussion section for the role of TCTP in cellular senescence and its possible relation with the pathophysiology of arterial hypertension (see Evangelou et al Physiol Rev 2022, Hu C, et al. Aging Dis. 2022).

Our response:

We are appreciating for your comments and suggestions about the potential pathophysiological relations between TCTP and the arterial hypertension in the view of cellular senescence. We have added a brief comment in the discussion section on the role of TCTP in cellular senescence and its potential association with the pathophysiology of arterial hypertension, as follows.

(Page 13, Line 562)

Further studies regarding hitherto unknown potential mechanisms and roles of TCTP in terms of its various cellular functions might delineate its pathophysiological network in the cardiovascular conditions. In this perspective, possible implications of TCTP in the cellular senescence and its contributory mechanism in the arterial hypertension and related disorders can be one of the future studies based on the differential roles of senescence in the majority of cardiovascular diseases [103,104].

  • Evangelou, K.; Vasileiou, P.V.; Papaspyropoulos, A.; Hazapis, O.; Petty, R.; Demaria, M.; Gorgoulis, V.G. Cellular Senescence and Cardiovascular Diseases: Moving to the “Heart” of the Problem. Rev. 2022, doi:10.1152/physrev.00007.2022.
  1. Hu, C.; Zhang, X.; Teng, T.; Ma, Z.-G.; Tang, Q.-Z. Cellular Senescence in Cardiovascular Diseases: A Systematic Review. Aging Dis. 2022, 13, 103–128, doi:10.14336/AD.2021.0927.

 Thank you very much for the comment and suggestion. We hope this revised manuscript will now be acceptable for publication in your journal.

We thank you in advance for your consideration.

Sincerely yours,

Prof. Kyunglim Lee

Graduate School of Pharmaceutical Sciences

College of Pharmacy

Ewha Womans University,

Seoul 03760, Republic of Korea

Tel: 82-2-3277-3024 

Fax: 82-2-3277-2851

Reviewer 2 Report

The manuscript entitled “Role of Translationally Controlled Tumor Protein (TCTP) in the Development of Hypertension and Related Diseases in Mouse Models” falls within the scope of the Journal. However, this reviewer has the following comments for the manuscript.

Minor comments:

 - Authors should insert a graphical abstract that summarizes the contents of the article in a concise form in order to capture the attention of the readership.

- Selected studies and inclusion and exclusion criteria are not clear. Authors should insert this section.

- Authors should insert an abbreviation section. The words for which is specified an abbreviation should be written in full the first time they are mentioned.

- Authors should include more recent literature and insert further adequate references in order to support their study.

- The English language has to be revised.

- Authors should improve the formal aspects of the manuscript.

Author Response

Dear Reviewer 2:

Thank you for providing me with the comments from reviewers of our manuscript entitled “Role of Translationally Controlled Tumor Protein (TCTP) in the Development of Hypertension and Related Diseases in Mouse Models”. We are now electronically submitting a revised manuscript, prepared taking into account the helpful critiques of the reviewers. We are grateful for this input, which we believe has strengthened the manuscript. We have provided below our detailed responses to the specific comments of the reviewers. These responses have also been included in the manuscript.

 Reviewer: 2

The manuscript entitled “Role of Translationally Controlled Tumor Protein (TCTP) in the Development of Hypertension and Related Diseases in Mouse Models” falls within the scope of the Journal. However, this reviewer has the following comments for the manuscript.

Minor comments:

1.      Authors should insert a graphical abstract that summarizes the contents of the article in a concise form in order to capture the attention of the readership.

 Our response:

As kindly commented by the reviewer, the graphical abstract file will be uploaded in the revision process, as follows.

(Graphical Abstract)

2. Selected studies and inclusion and exclusion criteria are not clear. Authors should insert this section.

Our response:

Thank you for your valuable comments. As suggested, we have supplemented the description on the criteria for the included and excluded studied in this review, as follows.

(Page 2, line 58) The following sentences were added.

Therefore, studies that examined the role of TCTP in the pathophysiology of arterial hypertension and potentially related conditions such as atherosclerosis, heart failure, and obesity, if any, were included for the description in the present review. In addition, studies that use genetically engineered mouse models whose TCTP expression was overexpressed or reduced either systemically or tissue-specifically, were included, whereas research that dealt with non-genetic models or genetic models unrelated to TCTP were excluded in this study.

3.      Authors should insert an abbreviation section. The words for which is specified an abbreviation should be written in full the first time they are mentioned.

 Our response:

As kindly mentioned, we checked the abbreviations in the manuscript and revised it, as follows.

(Page 1, Line 30) From ‘blood pressure’ to ‘blood pressure (BP)’

(Page 3, Line 117) From ‘blood pressure (BP)’ to ‘blood pressure’

(Page 2, Line 70) From ‘the FXYD family’ to ‘a Phe-X-Tyr-Asp (FXYD) motif-containing FXYD family’

(Page 3, Line 134) From ‘Ras-Raf-MEK-ERK 1/2 pathways’ to ‘Ras/Raf/mitogen-activated protein (MAP) kinase kinase (MEK)/extracellular signal-regulated kinase (ERK) 1/2 pathways’

(Page 5, Table 1)

From ‘p-MLC’ to ‘phospho-myosin light chain (p-MLC)’

From ‘FOXP3’ to ‘Forkhead box protein P3 (FOXP3)’

From ‘Tregs’ to ‘regulatory T cells (Tregs)’

From ‘Bnip3’ to ‘Bcl-2 interacting protein 3 (Bnip3)’

(Page 6, Line 215) From ‘ApoE’ to ‘apolipoprotein (ApoE)’

(Page 10, Line 398) From ‘LDL’ to ‘low-density lipoprotein (LDL)’

4.      Authors should include more recent literature and insert further adequate references in order to support their study.

We are thankful for your comment on the references. More recent literatures were supplemented to further support the present study in the revised manuscript, as follow.

(Page 2, Line 94)

Therefore, it was reported that mutations of sodium pump α1 is related to the development of secondary hypertension, an endocrine syndrome, a peripheral neuropathy, and neuromuscular disorders [15].

(Page 4, Line 182)

Interaction of sodium pump with TCTP is now generally accepted to have important roles not only in the biological functions and signaling but also in the pathophysiology of human diseases [34-36].

(Page 14, Line 562)

Further studies regarding hitherto unknown potential mechanisms and roles of TCTP in terms of its various cellular functions might delineate its pathophysiological network in the cardiovascular conditions. In this perspective, possible implications of TCTP in the cellular senescence and its contributory mechanism in the arterial hypertension and related disorders can be one of the future studies based on the differential roles of senescence in the majority of cardiovascular diseases [103,104].

(References)

  1. Biondo, E.D.; Spontarelli, K.; Ababioh, G.; Méndez, L.; Artigas, P. Diseases Caused by Mutations in the Na+/K+ Pump Α1 Gene ATP1A1. Am. J. Physiol. Cell Physiol. 2021, 321, C394–C408, doi:10.1152/ajpcell.00059.2021.
  2. Assrir, N.; Malard, F.; Lescop, E. Structural Insights into TCTP and Its Interactions with Ligands and Proteins. Results Probl. Cell Differ. 2017, 64, 9–46, doi:10.1007/978-3-319-67591-6_2.
  3. Bommer, U.-A. The Translational Controlled Tumour Protein TCTP: Biological Functions and Regulation. Results Probl. Cell Differ. 2017, 64, 69–126, doi:10.1007/978-3-319-67591-6_4.
  4. Lee, H.-J.; Song, K.-H.; Oh, S.J.; Kim, S.; Cho, E.; Kim, J.; Park, Y.G.; Lee, K.-M.; Yee, C.; Song, S.-H.; et al. Targeting TCTP Sensitizes Tumor to T Cell-Mediated Therapy by Reversing Immune-Refractory Phenotypes. Commun. 2022, 13, 2127, doi:10.1038/s41467-022-29611-y.
  5. Evangelou, K.; Vasileiou, P.V.; Papaspyropoulos, A.; Hazapis, O.; Petty, R.; Demaria, M.; Gorgoulis, V.G. Cellular Senescence and Cardiovascular Diseases: Moving to the “Heart” of the Problem. Rev. 2022, doi:10.1152/physrev.00007.2022.
  6. Hu, C.; Zhang, X.; Teng, T.; Ma, Z.-G.; Tang, Q.-Z. Cellular Senescence in Cardiovascular Diseases: A Systematic Review. Aging Dis. 2022, 13, 103–128, doi:10.14336/AD.2021.0927.

5.      The English language has to be revised.

We are thankful for your comment on the English language. The manuscript was under careful English editing and some of expressions were corrected in our revised manuscript, as follows.

(Page 1, Line 9) playing that plays

(Page 1, Line 10) roles, including

(Page 1, Line 15) increases / mice, leading

(Page 1, Line 31) leads to the damage to target organs, including

(Page 1, Line 34) heart failure, and chronic kidney disease (CKD).

(Page 1, Line 38) has long been

(Page 2, Line 58) the studies

(Page 2, Line 67) subunits, which appear

(Page 2, Line 72) The catalytic

(Page 2, Line 74) ouabain-binding site

(Page 2, Line 76) have been

(Page 3, Line 115) the sodium pump / that the inhibited

(Page 3, Line 120) the alteration

(Page 3, Line 131) α1 subunit, which is a

(Page 3, Line 142) contributes

(Page 4, Line 161) regulation both at transcriptional and translational levels

(Page 4, Line 166) atherosclerosis, and diabetes

(Page 4, Line 170) are potentially associated

(Page 4, Line 172) α1 and α2 isoforms

(Page 4, Line 181) question of whether

(Page 4, Line 192) the roles

(Page 5, Table 1) TCTP+/+ Ldlr-/- Apobec1-/-

(Page 6, Line 210) because, for example,

(Page 6, Line 214) TCTP-TG was

(Page 6, Line 215) with a hybrid background

(Page 7, Line 235) studied for further

(Page 7, Line 236) an increase in

(Page 7, Line 241) exhibit enhanced

(Page 7, Line 245) accompanied by

(Page 7, Line 261) the ouabain may

(Page 7, Line 262) has already been

(Page 7, Line 264) sensitivity to

(Page 7, Line 270) mobilization

(Page 7, Line 276) vasopressors in

(Page 7, Line 278) wild-type mice

(Page 7, Line 280) essential in inducing

(Page 8, Line 283) in a transgenic TCTP-overexpressing (TCTP-TG) mice.

(Page 8, Line 302, 304) myofilaments

(Page 8, Line 309) and it is

(Page 9, Line 328) underlies in the

(Page 9, Line 345) isolated from

(Page 9, Line 346) involved in

(Page 9, Line 350) There is a 2.13 times greater risk of having

(Page 9, Line 351) the 60-69 years

(Page 9, Line 353) cataracts was

(Page 10, Line 388) cholesterols in the lesions

(Page 10, Line 408) backgrounds

(Page 10, Line 412) an increase

(Page 11, Line 416) an exacerbation

(Page 11, Line 418) shows

(Page 11, Line 425) factors, including

(Page 11, Line 437) supports

(Page 11, Line 442) From ‘Pinkaew et al.’ to ‘Pinkaew et al.’

(Page 11, Line 443) TCTP-deficient

(Page 11, Line 460) more enhanced

(Page 12, Line 474) is positively

(Page 12, Line 487) the protective

(Page 12, Line 492) conditions, including

(Page 12, Line 497) the modifiable

(Page 12, Line 500) on metabolic

(Page 12, Line 502) wild-type

(Page 12, Line 511) the sympathetic

(Page 13, Line 537) confirms the protective

(Page 13, Line 553) TCTP could be

(Page 14, Line 572) help us understand

6.      Authors should improve the formal aspects of the manuscript.

As kindly commented on the formal aspect of the manuscript by the reviewer, we have corrected and revised the manuscript format, as it can be found in our revised manuscript.

Thank you very much for the comment and suggestion. We hope this revised manuscript will now be acceptable for publication in your journal.

We thank you in advance for your consideration.

Sincerely yours,

Prof. Kyunglim Lee

Graduate School of Pharmaceutical Sciences

College of Pharmacy

Ewha Womans University,

Seoul 03760, Republic of Korea

Tel: 82-2-3277-3024 

Fax: 82-2-3277-2851
